# "Pain in my heart": Understanding perinatal depression among women living with HIV in Malawi

Katherine LeMasters[1,2]*, Josée Dussault[1], Clare Barrington[2,3], Angela Bengtson[4], Bradley Gaynes[5], Vivian Go[3], Mina C. Hosseinipour[6,7], Kazione Kulisewa[8], Anna Kutengule[6], Samantha Meltzer-Brody[5], Dalitso Midiani[9], Steve Mphonda[6], Michael Udedi[8,9], Brian Pence[1]

1 Department of Epidemiology, Gillings School of Global Public Health, University of North Carolina at Chapel Hill, Chapel Hill, North Carolina, United States of America, 2 Carolina Population Center, Chapel Hill, North Carolina, United States of America, 3 Department of Health Behavior, Gillings School of Global Public Health, University of North Carolina at Chapel Hill, Chapel Hill, North Carolina, United States of America, 4 Department of Epidemiology, Brown School of Public Health, Brown University, Providence, Rhode Island, United States of America, 5 Department of Psychiatry, UNC School of Medicine, University of North Carolina at Chapel Hill, Chapel Hill, North Carolina, United States of America, 6 UNC-Project Malawi, Lilongwe, Malawi, 7 Department of Medicine, UNC School of Medicine, University of North Carolina at Chapel Hill, Chapel Hill, North Carolina, United States of America, 8 College of Medicine, University of Malawi, Lilongwe, Malawi, 9 Ministry of Health, Lilongwe, Malawi

* katherine.lemasters@unc.edu

## Abstract

### Background

Perinatal depression (PND) can interfere with HIV care engagement and outcomes. We examined experiences of PND among women living with HIV (WLWH) in Malawi.

### Methods

We screened 73 WLWH presenting for perinatal care in Lilongwe, Malawi using the Edinburgh Postnatal Depression Scale (EPDS). We conducted qualitative interviews with 24 women experiencing PND and analyzed data using inductive and deductive coding and narrative analysis.

### Results

Women experienced a double burden of physical and mental illness, expressed as pain in one's heart. Receiving an HIV diagnosis unexpectedly during antenatal care was a key contributor to developing PND. This development was influenced by stigmatization and social support.

### Conclusions

These findings highlight the need to recognize the mental health implications of routine screening for HIV and to routinely screen and treat PND among WLWH. Culturally appropriate mental health interventions are needed in settings with a high HIV burden.

**Data Availability Statement:** Data cannot be shared publicly because they contain sensitive patient information, as all women are HIV-positive and suffering from perinatal depression.

Furthermore, in our informed consent forms, it says "We will not share the information you give us with anyone not involved in the study." The ethics committee that imposes this the National Health Sciences Research Committee in Malawi, as we used their informed consent forms. The committee can be reached at: Tel: +265 1 726 422/418 or Email: mohdoccentre@nhsrc-mw.com.

**Funding:** This research was supported by grant R34 MH 116806 and R00 MH 112413 from the National Institute of Mental health (NIMH) and by a developmental grant from the UNC Center for AIDS Research (CFAR), an NIH-funded program (P30 AI 050410). This work was also supported by the Fogarty International Center through the Malawi HIV implementation research scientist training program (D43- D43TW010060). This paper does not reflect the views of the NIMH, NIH, or Fogarty.

**Competing interests:** The authors have declared that no competing interests exist.

## Introduction

The scale-up of antiretroviral therapy to all pregnant and breastfeeding women living with HIV, known as Option B+, has the potential to dramatically improve maternal health and end mother-to-child HIV transmission (MTCT) [1]. In Malawi, all pregnant women diagnosed with HIV in antenatal care (ANC) begin lifelong antiretroviral therapy (ART) under Option B + [2]. However, women who initiate ART during pregnancy under Option B+ are one-fifth as likely to return to HIV care after their initial visit compared to non-pregnant women initiating ART in Malawi [3]. In the short term, poor maternal mental health has the potential to undermine the delivery of Option B+ by affecting initiation of and retention in HIV care [4]. Long term, poor maternal mental health and disrupted HIV care may increase the risk of MTCT and have negative effects on women's quality of life and psychological well-being [4].

Globally, adults living with HIV are at an increased risk of depression, with the association being stronger among patients who are newly diagnosed and women [5]. A systematic review conducted in high-, middle-, and low-income countries found that pregnant and postpartum women living with HIV are at particularly high risk for perinatal depression (PND) due to multiple bio-psychosocial risk factors [4]. These risk factors include increased stress, HIV-related stigma, a lack of social support, concerns about disclosing their HIV status, and concerns about their infant's health and HIV status [4].

Through Option B+, more women are becoming aware of their HIV status and initiating ART during the perinatal period. Simultaneously, many are experiencing PND. PND is known to affect 13.1% of women in low and middle-income countries, with as many as 19.2% of women having a depressive episode within the first three months postpartum [6,7]. Among women living with HIV in Sub-Saharan Africa, a meta-analysis found a pooled prevalence for PND of 42.5% for prenatal women and 30.7% for postpartum women, indicating a high prevalence among this population [8].

PND is known to have detrimental effects on both mothers and infants [4,9]. For example, behavioral traits associated with depression (i.e., neglecting ANC) can lead to adverse effects on fetal health and child development [9]. Among women living with HIV, PND is also associated with increased risk for HIV progression as a function of dietary changes, impaired immune function, and suboptimal ART adherence and engagement in HIV care [10,11].

Given the connections between HIV, PND, and maternal and infant health, there is a great need for a fuller, qualitative understanding of the PND experience of women living with HIV in low-income settings [12,13]. Understanding the social etiology of PND will guide efforts to intervene on and alleviate PND among women living with HIV. Addressing PND among women living with HIV may also improve women's retention in HIV care, a global health priority, as well broader maternal and child health outcomes [14]. This study aims to understand the experience of PND among women living with HIV in Malawi.

## Methods

### Study site and population

We completed in-depth interviews about PND with women seeking pre- or postnatal care at five ANC clinics (two urban and three rural) in Lilongwe, Malawi between July and August 2018. All women living with HIV seeking pre- or postnatal care at the study sites who screened positive for PND and who were over the age of 18 were eligible for the study. HIV status was confirmed by women's health passports. PND was defined as depression occurring during pregnancy or the first 12 months postpartum [14,15], and was assessed using the Edinburgh Postnatal Depression Scale (EPDS), which was previously validated in Chichewa and is used to

screen for antenatal depression in this region [16–19]. Women were classified as having PND if they received a score of ≥ 10 on the EPDS. Consecutive women were screened by a trained counselor at each site until 24 women with PND total (4 at site A; 5 at sites B-E) were identified who agreed to participate in the study [20]. A sample size of 24 was decided upon to achieve data saturation [21]. Women reporting suicidal ideation in the EPDS were referred to mental health specialists as appropriate.

## Data instruments

We developed a semi-structured interview guide to explore women's experiences of PND, its determinants and manifestations, and its impact on HIV care engagement. This guide began by presenting PND symptoms and asking if they had seen someone with these symptoms and how they would describe them, then presented vignettes, or short stories with hypothetical characters, and closed by asking women about their personal experiences with depression. Vignettes were used due to the sensitivity of the study topic [22]. The vignette in the interview guide centered around a woman with a new child experiencing signs of PND and receiving an HIV diagnosis. The interviewer then asked how this woman would be treated in her community and how the woman being interviewed would handle the situation. The data collector then asked how the woman had been feeling in her most recent pregnancy and who she had confided in. The guide closed with a discussion of depression treatments, namely how the woman thought those experiencing depression would be most helped. The guide was created in English and translated into Chichewa. A trained, female research assistant from Malawi conducted all interviews and met with the study team weekly to discuss the data collection process.

## Analysis

All interviews were conducted and audiotaped in Chichewa, simultaneously transcribed and translated to English by AK, and uploaded to NVivo v.12 for data analysis [23]. We used a combination of thematic and narrative analysis [24]. Analysis and interpretation began during data collection as interviews were transcribed and translated [25]. After reviewing the first few transcripts, two research assistants (KL; JD) based in the United States created a codebook to begin categorizing data that included both descriptive and interpretive codes [24]. Using these descriptive and interpretive codes, the first author coded the data using a hybrid of data-driven (i.e., inductive) coding and concept-driven (i.e., deductive) coding, with concepts coming from prior literature, the research team's previous experience, and the research questions [26]. The first author also analyzed words and phrases that were significant, listed their meanings, and created *in vivo* codes to capture phrases women used to describe their PND [13,26]. The first author met regularly with the Malawian interviewer and translator and other members of the research team to clarify when translation was unclear, refine the codebook, and identify themes.

The analysis also involved creating analytic memos to document the development of the team's understanding of women's experiences of living with both PND and HIV and to identify emergent patterns and relationships between codes, which assisted in connecting the data [26,27]. The first author then created matrices to identify and analyze similarities and differences between participants for key themes [28]. Lastly, using narrative analysis, the first author wrote an HIV diagnosis narrative based on the participant 'Ruth.' Because women's stories often began with their HIV-diagnosis, a life-changing event, the narrative structure assisted in analysis by establishing chronology [24,29,30]. We use the illustrative case of Ruth to highlight

processes over time within one woman's experience, as a complement to thematic summaries and quotes.

### Ethics

This study was approved by the Institutional Review Boards at the University of North Carolina at Chapel Hill (17–2396) and at the Malawi National Health Sciences Research Committee. If participants were literate, written consent was obtained. If participants were illiterate, oral consent was obtained and an impartial witness was present.

## Results

### Demographic information

73 women were screened and 24 (33%) had elevated symptoms of PND. We conducted 24 in-depth interviews (14 with prenatal and 10 with postpartum women with living with HIV and PND). Of the 24 with elevated symptoms of PND, 14 were pregnant and 10 were postpartum at the time of the interview (Table 1). The proportion of women with PND was higher at the three rural sites (range: 45–71%) compared to the two urban sites (13–29%) (Table 1). Of the 73 women screened, 14 (19% of all women; 58% of those with PND) reported suicidal ideation. Report of suicidal ideation was also higher at the three rural sites (27–63% of all women) than at the urban sites (0–24% of all women).

Women were, on average, 27 years old and most had more than one child, were married, unemployed, and had at least some primary education (Table 2). One woman was beginning ART at the current appointment while the remainder had already initiated ART. Most women (71%) had received their HIV status over two years ago and none had been screened for or diagnosed with PND previously.

### Women's experience of PND

Here we present the experiences of women living with both HIV and PND in Malawi. We highlight the narrative of one respondent, 'Ruth,' alongside others' stories to demonstrate how PND often manifests and how a woman's HIV diagnosis is a key contributor to her development of PND. Ruth's story represents a typical case, which helps provide insight into women's PND experiences.

**"Pain in my heart": A double burden.** Ruth was a pregnant participant who had received an HIV diagnosis during a past pregnancy over two years ago. When Ruth and others described how their PND presented itself in their lives, they often described their culmination of symptoms causing "pain in my heart." When talking about her depression during her

**Table 1. Perinatal depression (PND) screening.**

| Clinic Site | Clinic Locale | Number Screened for PND+ | PND | Suicidal Ideation |
|---|---|---|---|---|
| | | | N (%) | N(%) |
| Site A | Urban | 30 | 4 (13) | 0 (0) |
| Site B | Urban | 17 | 5 (29) | 4 (24) |
| Site C | Rural | 11 | 5 (45) | 3 (27) |
| Site D | Rural | 7 | 5 (71) | 2 (29) |
| Site E | Rural | 8 | 5 (63) | 5 (63) |
| Total | | 73 | 24 (33) | 14 (19) |

[+] Women were screened until 4 to 5 were found to have PND at each site.

**Table 2. Women's demographic information.**

| Women's Demographic Information | N (%) |
|---|---|
| Type of Participant | |
| Prenatal Woman | 14 (58%) |
| Postnatal Woman | 10 (42%) |
| Age (Mean (SD)) | 27 (5.48) |
| Pregnancy Number | |
| First | 3 (13%) |
| Second | 9 (38%) |
| Third | 6 (33%) |
| Fourth | 4 (17%) |
| Fifth or more | 2 (8%) |
| Marital Status | |
| Married | 20 (83%) |
| Separated | 2 (8%) |
| Divorced | 2 (8%) |
| Education | |
| None | 2 (8%) |
| S1-S7 | 12 (50%) |
| S8 | 2 (8%) |
| Secondary or More | 8 (33%) |
| Initiating ART for the first time | |
| Yes | 1 (4%) |
| Length Living with HIV | |
| 2+ years | 17 (71%) |
| 1–2 years | 1 (4%) |
| 6 months– 1 year | 1 (4%) |
| Diagnosed in last 6 months | 5 (21%) |

interview, Ruth said that her heart was troubled; her depression would not stop in her heart and was persisting because it had come from the combination of her HIV diagnosis, pregnancy, and marital issues. Other women used phrases referring to the heart. Women used this phrase of having pain in their hearts to describe how depression felt to them and how this feeling persisted in their heart throughout time, disrupting their lives and not allowing them to tend to other tasks. When a prenatal woman that had been diagnosed in her current pregnancy was asked if she knew about depression, she described her own experience as the following: "from the time I started my HIV treatment, I feel a lot of pain in my heart. It's not like I am worried about anything, but I just feel so much pain in my heart, as if I have been shocked by something" (prenatal, living with HIV <6 months). This pain persisted because she did not feel that she could confide in anyone about her HIV status and because her husband had abandoned her once she disclosed her diagnosis. A woman diagnosed with HIV within the last two years said that she kept feeling an overwhelming pain in her heart that kept her from working and that this pain stemmed from her overthinking and worry. This worry was about raising her children alone, as her husband had abandoned her after she became HIV-positive. Importantly, women explained that understanding how others felt in their heart and helping other women strengthen their heart were potential mechanisms for addressing PND.

The persistence of women's depression was often expressed through women's double burden of living with both HIV and PND. This double burden was closely tied to having an

unexpected HIV diagnosis, with the diagnosis making the depression harder to handle. During her interview, Ruth described the persistence of her depression and the pain in her heart as a large burden. Discussions of this double burden were most common when women were asked to imagine a woman with PND and HIV and one with only PND. HIV and PND were both thought of as diseases, one being physical and one being mental: "one of them [with only PND] is just depressed but her body is okay whereas the other one [with both PND and HIV] is depressed and her body has viruses" (prenatal, living with HIV over 2 years). All women said that living with both would be more difficult and would be different than only having PND.

Women also expressed that they could handle their PND more easily if it did not co-occur with HIV. As a woman's HIV diagnosis was a key contributor to her developing PND, without this diagnosis, many women believed they would not have developed it or would have experienced a milder form. One woman stated that receiving an HIV diagnosis made her PND worse: "you get very depressed because you think of the problems you already have at home and then you have even more problems now. The depression increases" (postnatal, living with HIV over 2 years). Without an HIV diagnosis, PND was also perceived to have an endpoint whereas PND stemming from HIV was expected to be lifelong because the primary cause (i.e., their HIV status) was also lifelong.

Ruth and others felt that this double burden combined with stigma and marital issues led them to have suicidal thoughts, as they claimed that committing suicide would rid them of all of their problems. Fourteen out of 24 women (58%) expressed suicidal ideation and such thoughts were more common among currently pregnant women and those lacking support from their partner. Of the 14 women with suicidal thoughts, nine had passive or low-risk suicidal ideation and five had active or high-risk suicidal ideation.

In addition to suicidal thoughts, women also revealed that their burdens could affect their ART adherence. Ruth explained that "if you are depressed, you cannot manage to do those things [take ART medication] because of the depression and you're hurt in your heart every day, because when someone is depressed the heart always hurts. For you to be bothered about your life, you just say whatever happens will happen." Women's PND resulted in hopelessness, which could lead them to either forget to take their medication or to lack the motivation to go to the clinic. At the same time, Ruth said that women often think too much when they have PND, and it is harder to remember one's medication when one has so many thoughts.

Yet, suicidal thoughts did not result in suicide attempts and ART barriers did not result in women's lack of adherence to ART for Ruth and many others. High reported adherence was most common among women in urban areas and among postnatal women. Ruth was motivated to take her ART "so that the baby [she was] expecting should not go through the burden that [she was] going through" by contracting HIV. She also felt responsible for her other children and wanted to remain healthy so that she could raise them. Adherence was often made easier when women had disclosed their status to others, as they then felt accountable to those to whom they had disclosed and felt encouraged to accept their HIV status and begin and remain in treatment.

**Intersecting identities: HIV-positive, pregnant, and in an unstable marriage.** Ruth had known she would be tested for HIV as part of routine ANC, but did not suspect that she had contracted HIV. In discussing her HIV diagnosis, she stated that "depression is inevitable because you have been diagnosed with HIV at a time you were not expecting it." Ruth found it difficult to accept the reality of having HIV when she did not anticipate a diagnosis.

For Ruth and others that received their HIV diagnosis during pregnancy, their diagnosis compounded pre-existing anxieties about being a mother, predisposing them to develop PND. One woman in her fourth pregnancy began "wondering how [she] would be able to look after

[her children] with the HIV and [kept] wondering if this was the end of [her] life" after her diagnosis (postnatal, living with HIV 1–2 years). She continually expressed her worry about her diagnosis and about how she would be able to parent while living with HIV.

In addition to being pregnant when receiving her diagnosis, Ruth learned that her husband was also HIV-positive but had been hiding his diagnosis from her. Upon learning of Ruth's diagnosis, her husband acted as though she "went wayward and brought the virus into the marriage." Like Ruth, most women were in relationships in which they believed they had contracted HIV from their husband. Yet, most women's husbands reacted negatively to their wives' HIV diagnosis and denied their own responsibility. Women's HIV diagnoses combined with their pregnancy often exacerbated preexisting marital issues and created new ones. These marital issues meant that women lacked social support from a partner, contributing to them both developing PND and accelerating and exacerbating it:

"I sometimes do not talk to him because something is troubling me. To think the same person who transmitted the virus to me is the one who is insulting me. . ."

(prenatal, living with HIV < 6 months).

For Ruth, the combination of not anticipating her HIV diagnosis, being pregnant when she received her diagnosis, and unknowingly contracting HIV from her husband resulted in her feeling overwhelmed and like her challenges were insurmountable. It was then difficult for her to accept the reality of having HIV in the midst of her overwhelming circumstances. Many other women noted an inability to accept an HIV diagnosis or a denial of their diagnosis as contributing to feeling depressed. While an inability to accept a diagnosis and an active denial of a diagnosis may be different, women used these two descriptions interchangeably.

**Stigma and social support.** In addition to their marital issues, many women felt stigmatized and unsupported by others. One woman stated that "people who can encourage us to live a life with little depression are rare" (prenatal, living with HIV over 2 years). Both HIV and PND were emotionally charged topics in women's communities and women often received mixed reactions from community members. Mixed or negative reactions largely stemmed from others not accepting women's HIV diagnosis, meaning that they did not accept the women as their full selves with an HIV diagnosis. One woman directly linked HIV stigma to her development of depression: "I sometimes ask myself if [my community knows] about my HIV status and if that is the reason they treat me in the way they do. I then get depressed because I think too much" (postnatal, living with HIV over 2 years). Women feeling stigmatized by others often led to their overthinking, which was both a cause and symptom of PND.

While HIV stigma was a prominent determinant of some women's depression, others felt directly stigmatized for their depression because they felt that they were perceived as a bad mother. Women were sometimes warned that they "shouldn't be sad, because the baby [will] also be sad" (prenatal, living with HIV over 2 years), or were described as sick, mad, in trouble, lazy, or panicked. Importantly, once women were stigmatized due to their PND, they sometimes began to worry that people were stigmatizing them due to their HIV status even if people did not know their status, which created an internal cycle of worry. Still others felt a lack of love or ambivalence towards them once people knew about their HIV or PND. Ambivalence towards women in these circumstances was often as negative as active stigmatization, as women were not supported and thus unable to move towards accepting their status and lessening their PND.

Yet, some women found sources of support. Linked to their denial or struggle to come to terms with their HIV status, women did not immediately disclose their HIV status or feelings

of depression to others. Rather, they turned to prayer. Fourteen of the 24 women listed prayer as a source of support in response to their HIV diagnosis. One woman said that she "would just pray for God to remove [her] worries because. . .He is the one who can remove [her] anxieties" (prenatal, living with HIV over 2 years), indicating that both she and those around her could not remove her anxieties related to her HIV.

After prayer, many women turned to one or two specific people for support. Ruth confided in one of her in-laws, and most women talked to a family member, friend, or their husband. While most did not find support in their husbands, four women found encouragement in talking with them, including two women who noted that their husbands were HIV-positive. Women that had been living with HIV for longer were more likely to have found sources of support. Additionally, if women knew others that were living with HIV, they were likely to serve as sources of support.

Throughout women's discussions of social support, they described social support as being composed of three components: interaction, encouragement, and offloading or sharing worries. Interaction was seen as a distraction for women to stop worrying about their HIV and PND. Encouragement was discussed as both needed by women and provided by them to others so that "[they] get strengthened in [their] heart" (prenatal, living with HIV over 2 years). Encouragement thus helped their depression and their worries. Offloading was identified as a critical component for preventing suicide. Some had not found anyone that they could share their worries with: "I have never met anyone whom I could share my worries with. . . sometimes I get depressed but can't tell anyone" (postnatal, living with HIV over 2 years). Thus, women needed a combination of all three components of social support, as they all served different functions in alleviating PND.

## Discussion

In our population, women with PND experienced a unique double burden of HIV and PND, which was commonly expressed as having pain in their hearts and as worry. Women's HIV diagnosis, especially when it was unexpected and received during routine ANC, was a key contributor to their developing PND. Women's unexpected diagnosis intersected with their pregnancy and marital relationships to contribute to their PND. These relationships were then influenced, positively or negatively, by women's social interactions and relationships.

Our study sheds light on the experience of women living with both PND and HIV in a low-income country with high HIV prevalence. Three main themes emerged from our interviews: a double burden of having a physical and mental illness, as expressed through having pain in one's heart; women's intersecting identities of being HIV-positive, pregnant, and in an unstable marriage; and the key roles of stigmatization and social support (or lack thereof) in influencing the development and trajectory of PND.

First, given the many contributing factors to women's PND, it is not surprising that women would experience a large burden. Yet, while the co-occurrence of HIV and depression is often cited in the literature about this population, there is a lack of discussion around what this co-occurrence means for women's experiences and the burden it creates [4,31–33]. This finding reemphasizes that these two epidemics often collide in the lives of women and their intersection deserves global attention through increased screening and treatment [14].

Second, the high interconnection of contributors to PND is supported in other qualitative work [13]. Women's HIV diagnosis has been noted as a source of depression in prior literature in sub-Saharan Africa, with women reporting a 3.5-fold higher number of mental health issues, especially depression, after an HIV diagnosis [31,32]. We add to this literature by finding this HIV diagnosis to be particularly burdensome when it is not expected and is diagnosed during

routine ANC. Research from South Africa found that if women are living with HIV, women's PND is worse because women are worried about having children to look after while living with HIV [13]. The role of a woman's marriage was quite prominent in our data. Based on our data, it is possible that women living with HIV and experiencing PND may, in particular, lack support from their partner. Another study in Lilongwe, Malawi found that women that had not disclosed their HIV status to their partner had twice the prevalence of PND, which likely indicates a poor relationship with and emotional support from a partner [13,34]. Thus, women's marriage may relate to their depression specifically among HIV-positive women, as HIV women's HIV status may indicate a lack of ability to have previously negotiated condom use with a partner and a lack of emotional support within marriage [35].

Third, women's stigma related to HIV and PND cut across all of their identities mentioned above [35,36]. Most prior work focuses on HIV-related stigma specifically and finds that women reporting greater stigma related to HIV in Malawi are significantly more likely to report depression [36]. This indicates that in addition to women having an unexpected HIV diagnosis, HIV may contribute to PND through stigma. A study in South Africa found that the relationship between stigma and HIV among HIV-positive women persists after controlling for marital status and pregnancy intention, indicating that it is a distinctly important component of women's identities contributing to PND [37]. Stigma related to depression plays a role as well. One study in South Africa found that psychological or emotional illnesses, including depression, hold an additional layer of stigma and increase stress and perceived stigma upon disclosure of status [38]. Yet, this work was not specific to PND [38]. While we found depression-related stigma to be present, we argue that there is unique stigma related to PND, as women were worried about being perceived as a bad mother due to their depression. Additionally, while the literature documents well how stigma affects depression, it is less well understood how ambivalence or a lack of support can be a contributing factor. Multiple women did not mention overt stigma, but indicated that ambivalence and a lack of support also contributed to their PND.

Regarding the association between PND and HIV, most literature reports that depression is associated with lower ART adherence, which often serves as the motivation for why PND should be addressed among HIV-positive women. Yet, while the women in our study said that other depressed women may not adhere to ART, most claimed that they did not have issues with adherence themselves. It is possible that the women choosing to participate in our interviews were more likely to be those well engaged in care; it is also possible the women felt some social desirability pressure to report good adherence for themselves, while feeling more free to note that others might have such a difficulty. One explanation, as noted by another study in Lilongwe, Malawi is that because these women are in a population with high engagement in care, PND is not associated with lower ART adherence [39]. Another explanation our study found is that during this perinatal period, women may be more motivated to adhere to ART as they want to have a healthy pregnancy and remain healthy for raising their children. It is important to note this motivating factor, as it may be protective in women's ART retention during the perinatal period.

## Recommendations

Moving forward, literature suggesting that HIV and depression care should be combined in treatment and that providers should attend to the emotional and psychological needs of women need to be transferred to the perinatal population and expand beyond ART counseling [32,36]. Additionally, examples exist that address PND at the community level using pre-existing primary care structures and training lay health workers in counseling [40–42]. Given that

HIV-infected women often experience a double burden of physical and mental illness that is influenced by social support, stigmatization, and family dynamics, these interventions have the possibility of being expanded and adapted to women living with HIV. Specifically, the Friendship Bench, individual talking therapy based on problem-solving therapy delivered by a lay health worker, has been found to be effective in Zimbabwe among a general population suffering from depression, the majority of whom were living with HIV [41]. Importantly, the Friendship Bench and similar interventions draw on pre-existing structures of primary care and ART counseling services and train lay health workers, indicating that they are scalable given the high proportion of women living with HIV that also have PND. Additionally, they focus on counseling and providing social support, which is often a critically missing piece of women's current experiences.

## Limitations

Our findings should be considered in the context of certain limitations. First, while the use of vignettes is an appropriate strategy for discussing sensitive topics, it creates difficulty in disentangling women's personal experiences from their perceptions of others' experiences. Second, the data was translated directly into English from an audio-recording in Chichewa, which may result in inconsistencies in translation. However, the translator was an active member of the research team and was available for discussion, which aided immensely in data analysis and interpretation [43]. Third, it is possible that there was social desirability bias in discussing ART adherence. Lastly, we have limited generalizability, as we only spoke with women engaged in ART and with women that were willing to talk about their PND from five clinics in Malawi. However, our five sites do capture a diverse patient population and all women that screened positively for PND agreed to be interviewed.

## Conclusions

By improving our understanding of the social etiology of PND among women living with HIV, we will be better able to construct interventions that are specifically designed for women living with both PND and HIV and that are responsive to the experiences of women [13]. In conclusion, our findings indicate a great need for programs to recognize and address the mental health issues during routine HIV testing and ART treatment and the importance of recognizing women's whole identities and experiences in assessing the burden of PND in this population.

## Supporting information

**S1 Data.**
(DOC)

**S2 Data.**
(DOCX)

## Acknowledgments

The authors would like to thank the team at UNC Project Malawi. We are deeply grateful to the women that participated in this project.

## Author Contributions

**Conceptualization:** Katherine LeMasters, Angela Bengtson, Bradley Gaynes, Vivian Go, Mina C. Hosseinipour, Kazione Kulisewa, Samantha Meltzer-Brody, Dalitso Midiani, Steve Mphonda, Michael Udedi, Brian Pence.

**Data curation:** Anna Kutengule.

**Formal analysis:** Katherine LeMasters, Josée Dussault, Clare Barrington, Brian Pence.

**Funding acquisition:** Angela Bengtson, Bradley Gaynes, Mina C. Hosseinipour, Kazione Kulisewa, Samantha Meltzer-Brody, Dalitso Midiani, Steve Mphonda, Michael Udedi, Brian Pence.

**Methodology:** Clare Barrington, Angela Bengtson, Bradley Gaynes, Kazione Kulisewa, Anna Kutengule, Brian Pence.

**Project administration:** Angela Bengtson, Mina C. Hosseinipour, Dalitso Midiani, Steve Mphonda, Brian Pence.

**Resources:** Steve Mphonda.

**Supervision:** Clare Barrington, Angela Bengtson, Bradley Gaynes, Vivian Go, Mina C. Hosseinipour, Kazione Kulisewa, Samantha Meltzer-Brody, Dalitso Midiani, Steve Mphonda, Michael Udedi, Brian Pence.

**Writing – original draft:** Katherine LeMasters, Josée Dussault, Clare Barrington.

**Writing – review & editing:** Katherine LeMasters, Josée Dussault, Clare Barrington, Angela Bengtson, Vivian Go, Mina C. Hosseinipour, Kazione Kulisewa, Anna Kutengule, Samantha Meltzer-Brody, Dalitso Midiani, Steve Mphonda, Brian Pence.

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
