## [Decision Letter · Decision Letter 0]

24 Mar 2020

PONE-D-20-00046

"Pain in my heart": Understanding perinatal depression among women living with HIV in Malawi

PLOS ONE

Dear Dr. LeMasters,

Thank you for submitting your manuscript to PLOS ONE. After careful consideration, we feel that it has merit but does not fully meet PLOS ONE’s publication criteria as it currently stands. Therefore, we invite you to submit a revised version of the manuscript that addresses the points raised during the review process.

We would appreciate receiving your revised manuscript by May 08 2020 11:59PM. To enhance the reproducibility of your results, we recommend that if applicable you deposit your laboratory protocols in protocols.io, where a protocol can be assigned its own identifier (DOI) such that it can be cited independently in the future. For instructions see: http://journals.plos.org/plosone/s/submission-guidelines#loc-laboratory-protocols

We look forward to receiving your revised manuscript.

Kind regards,

Ali Montazeri

Academic Editor

PLOS ONE

Journal Requirements:

2. Please include a copy of the interview guide used in the study, in both the original language and English, as Supporting Information, or include a citation if it has been published previously.

"This research was supported by grant R34 MH 116806 and R00 MH 112413 from the National Institute of Mental health (NIMH) and by a developmental grant from the UNC Center for AIDS Research (CFAR), an NIH-funded program (P30 AI 050410). This paper does not reflect the views of the NIMH or NIH."

5. Your ethics statement must appear in the Methods section of your manuscript. If your ethics statement is written in any section besides the Methods, please move it to the Methods section and delete it from any other section. Please also ensure that your ethics statement is included in your manuscript, as the ethics section of your online submission will not be published alongside your manuscript.

Reviewers' comments:

Reviewer's Responses to Questions

**Comments to the Author**

1. Is the manuscript technically sound, and do the data support the conclusions?

Reviewer #1: Yes

Reviewer #2: Yes

2. Has the statistical analysis been performed appropriately and rigorously? 

Reviewer #1: Yes

Reviewer #2: Yes

3. Have the authors made all data underlying the findings in their manuscript fully available?

Reviewer #1: Yes

Reviewer #2: Yes

4. Is the manuscript presented in an intelligible fashion and written in standard English?

Reviewer #1: Yes

Reviewer #2: Yes

5. Review Comments to the Author

Reviewer #1: PONE-D-20-00046

"Pain in my heart": Understanding perinatal depression among women living with HIV in Malawi.

The comments for the above manuscript:

Short Title:

Short title is just like the main title so it must be corrected.

Abstract:

- In the Methods: The research method is not mentioned.

Introduction:

- The beginning sentences for the introduction seems to be not suitable because the HIV treatment is not the main problem of the present study.

- Line 45-48: It is not clear, what does it mean?

- The authors have noted that the disease is associated with depression, so the need for a study is unclear. They have mentioned to “women living with HIV in low income settings” but there is not any explanation about the reasons for probable differences of the PND in this group of women with others that previous studies have conducted on them. So the necessity of the present study should be clarified.

Methods:

- How many women completed the EPDS questionnaire? What was the sampling method?

- EPDS questionnaire is a postnatal depression scale. How the authors explain for using it during pregnancy?

- What was the evidence for living with HIV?

Results:

- Is written in a pleasant and expressive way.

- Table 1: It is better to write the percentages in the last two columns next to their related numbers (4th and 5th column) in the parenthesis.

Discussion:

- Is well written.

Reviewer #2: This article is technically and conceptually well organized. All the criteria of a research article is considered in writing.

I suggest some revision in line 85 as it is not clear whether 4 or 5 women were interviewed.

Also, I think it would be better to assign the statistical tests used in table 1 in lines 132-136.

6. PLOS authors have the option to publish the peer review history of their article (what does this mean?). If published, this will include your full peer review and any attached files.

Reviewer #1: No

Reviewer #2: No

---

## [Author Response · Author response to Decision Letter 0]

14 Apr 2020

Response to Reviewers

We very much appreciate the time taken to review our manuscript and for your thoughtful comments.

Reviewer #1: 

Short Title: 

Short title is just like the main title so it must be corrected.

- Thank you for this, it is now “Perinatal depression and HIV in Malawian Women.”

Abstract:

In the Methods: The research method is not mentioned.

- Thank you for this. We have only included a brief description of our methods in the abstract due to word limitations, but we have been sure to include our sample size and that we used both inductive and deductive coding and narrative analysis (page 2 lines 27-29). I have also added ‘qualitative’ in front of ‘interviews’ for further clarification. 

Introduction:

The beginning sentences for the introduction seems to be not suitable because the HIV treatment is not the main problem of the present study.

- Thank you for your comment. While it is true that HIV treatment is not the only problem of this study, the statement is meant to provide a broad overview of how HIV treatment and maternal mental health intersect and may contribute to poor HIV outcomes. We hope that our revisions to what were lines 45-48 below help to clarify this. 

Line 45-48: It is not clear, what does it mean?

- Thank you for your question. We have edited this statement, which is meant to connect poor HIV retention in care with maternal mental health (page 2 line 47-50), as our study explores the connection between the two. 

The authors have noted that the disease is associated with depression, so the need for a study is unclear. They have mentioned to “women living with HIV in low income settings” but there is not any explanation about the reasons for probable differences of the PND in this group of women with others that previous studies have conducted on them. So the necessity of the present study should be clarified.

- Thank you for your comment. As you said, and as stated in the manuscript, we know that perinatal depression affects many women in low and middle-income countries and we know that adults living with HIV are at an increased risk for depression. However, very little is known about perinatal depression among women living with HIV. Indeed, what we do know is quantitative and/or is not specific to only women living with HIV (added to page 3 line 77).1,2 Thus, there is a need to develop a richer, qualitative understanding of PND specifically among HIV-positive women. Additionally, mental illness and its stigma is conceptualized differently in different settings, further indicating the need to deeply explore its intersection with HIV in various settings.3 

Methods:

How many women completed the EPDS questionnaire? What was the sampling method?

- Under Results, Demographic Information (page 6, line 145), it states that 73 women were screened. This is also noted in the Abstract. 

- Page 4, lines 93-95 states that women were sampled consecutively at each site until the desired sample size was reached. Consecutive sampling is very similar to convenience sampling. Each eligible patient who presents for care is approached for enrollment until the desired sample size has been reached.4 The citation has now been added to the manuscript. 

EPDS questionnaire is a postnatal depression scale. How the authors explain for using it during pregnancy?

- Thank you for your question. While originally designed to screen for postpartum depression, the EPDS is frequently used to screen for depression in antenatal settings, including in sub-Saharan Africa, which is now included in the manuscript (page 4, lines 91-92).5,6

What was the evidence for living with HIV?

- Women’s health passports were used to determine their HIV status, this has been updated in the manuscript in the methods section (page 4, lines 88-89).

Results:

Is written in a pleasant and expressive way.

- Thank you for your feedback, it’s greatly appreciated.

Table 1: It is better to write the percentages in the last two columns next to their related numbers (4th and 5th column) in the parenthesis.

- Thank you for this suggestion, this has been done and Table 1 reflects these changes (page 7).

Discussion:

Is well written.

- Thank you so much for your comments.

Reviewer #2: 

This article is technically and conceptually well organized. All the criteria of a research article is considered in writing.

- Thank you very much for your review and comments.

I suggest some revision in line 85 as it is not clear whether 4 or 5 women were interviewed.

Also, I think it would be better to assign the statistical tests used in table 1 in lines 132-136.

- Thank you for your comment. 24 women were recruited, requiring one site to have 4 women while the remaining sites recruited 5. This has been added to the manuscript (page 4, lines 94-95). Our goal in qualitative interviews was to reach saturation, where new data are redundant with data already collected.7 Given the rule of thumb of 5-10 in-depth interviews per stratum (pre- and post-natal), we expected that we would achieve saturation within our proposed sample size. This has also been added to the manuscript (page 4, lines 128-129). We hope that this change clarifies the second part of your statement, as we are not sure what the statisticial tests are that you are referencing.

---

## [Editor Report · Decision Letter 1]

26 May 2020

"Pain in my heart": Understanding perinatal depression among women living with HIV in Malawi

PONE-D-20-00046R1

Dear Dr. LeMasters,

We are pleased to inform you that your manuscript has been judged scientifically suitable for publication and will be formally accepted for publication once it complies with all outstanding technical requirements.

With kind regards,

Ali Montazeri

Academic Editor

PLOS ONE
---

## [Editor Report · Acceptance letter]

28 May 2020

PONE-D-20-00046R1 

"Pain in my heart": Understanding perinatal depression among women living with HIV in Malawi 

Dear Dr. LeMasters:

I am pleased to inform you that your manuscript has been deemed suitable for publication in PLOS ONE. Congratulations! Your manuscript is now with our production department. 

With kind regards,

on behalf of

Professor Ali Montazeri 

Academic Editor

PLOS ONE